# Seasonal effects of irrigation on land-atmosphere latent heat, sensible heat and carbon fluxes in semi-arid basin

Yujin Zeng[1, 2], Zhenghui Xie[1], Shuang Liu[1, 2]

[1]State Key Laboratory of Numerical Modeling for Atmospheric Sciences and Geophysical Fluid Dynamics, Institute of Atmospheric Physics, Chinese Academy of Sciences, Beijing, 100029, China
[2]College of Earth Science, University of Chinese Academy of Sciences, Beijing 100049, China

*Correspondence to*: Zhenghui Xie (zxie@lasg.iap.ac.cn)

**Abstract.** Irrigation, which constitutes ~70% of the total amount of fresh water consumed by the human population, is significantly impacting the land-atmosphere fluxes. In this study, using the improved Community Land Model version 4.5 (CLM 4.5) with an active crop model, two high resolution (~1 km) simulations investigating the effects of irrigation on Latent Heat (LH), Sensible Heat (SH) and Carbon Fluxes (or net ecosystem exchange, NEE) from land to atmosphere on the Heihe River Basin in northwestern China were conducted using a high-quality irrigation dataset compiled from 1981 to 2013. The model output and measurements from remote sensing demonstrated the capacity of the developed models to reproduce ecological and hydrological processes. The results revealed that the effects of irrigation on LH and SH are strongest during summer with a LH increase of ~100 W m$^{-2}$ and a SH decrease of ~60 W m$^{-2}$ over intensely irrigated areas. However, the reactions are much weaker during spring and autumn when there is much less irrigation. When the irrigation rate is below 5 mm day$^{-1}$, the LH generally increases, whereas the SH decreases with growing irrigation rates. However, when the irrigation threshold is in excess of 5 mm day$^{-1}$, there is no accrued effect of irrigation on the LH and SH. Irrigation produces opposite effects to the NEE during spring and summer. During the spring, irrigation yields more discharged carbon from the land to the atmosphere, increasing the NEE value by 0.4–0.8 gC m$^{-2}$ day$^{-1}$, while the summer irrigation favors crop fixing of carbon from atmospheric $CO_2$, decreasing the NEE value by ~0.8 gC m$^{-2}$ day$^{-1}$. The repercussions of irrigation on land-atmosphere fluxes are not solely linked to the irrigation amount, and other parameters (especially the temperature) also control the effects of irrigation on LH, SH and NEE.

# 1 Introduction

Irrigation consumes most of the fresh water exploited by the global human population (Douglas et al., 2006). According to statistics obtained from the Food and Agriculture Organization (FAO) of the United Nations, about 2,722 billion $m^3$ or 70% of the total global anthropogenic water extraction was consumed by agricultural irrigation in 2007. In Asia and Africa, where agriculture contributes the most to the Gross Domestic Product (GDP), irrigation becomes more important and consumes more than 80% of total exploited fresh water (Oweis and Hachum, 2006).

Anthropogenic carbon emissions from fossil fuel combustion have already primarily changed the concentration of carbon dioxide in the atmosphere leading to an increase in global temperatures (Cox et al., 2000; Bindoff et al., 2013). Decades of large-scale and intensive farmland irrigation may also have greatly influenced the land-atmosphere water budget (Vörösmarty and Sahagian, 2000). Theoretically, all major land to atmosphere fluxes are directly or indirectly impacted by irrigation (Chen and Xie, 2010). Irrigation water can moisten the soil root-zones and alleviates the water stress on crops, generating more water consumption by ground evaporation, vegetation evaporation and crop transpiration (collectively called evapotranspiration, or ET) to be returned to the atmosphere (Kendig et al., 2000; Yu et al., 2014). This results in a direct augmentation of the latent heat flux (LH) from land to atmosphere (Haddeland et al., 2006). Besides, the irrigation changes the partitioning of net radiation available at the surface into LH and SH (Bowen ratio) (Meijninger and De Bruin, 2000). Vertical carbon fluxes are also affected. Submitted to a lesser water stress related to irrigation, crops grow more vigorously and sequester more carbon from atmospheric $CO_2$ during the growth season (Vitousek et al., 1997), inducing a negative change in Carbon Flux (also called net ecosystem exchange, or NEE in ecosystem ecology) between land and atmosphere. The NEE variation is very important for it is directly linked to crop yields. In a larger perspective, the NEE is related to greenhouse gases and global warming (Xie et al., 2014). The contribution of agricultural irrigation to other land-atmosphere fluxes, such as the Momentum Fluxes closely related to crop height, the Nitrogen and Oxygen Fluxes accompanying the NEE, has also been demonstrated by previous studies (Forster and Graf, 1995; Scheer et al. 2008). When vertical fluxes are modified, other atmospheric variables (e.g. air temperature, precipitation, carbon-nitrogen concentration, etc.) may vary correspondingly at the local and regional scales, and even be globally changed through atmospheric circulation (Chen and Xie, 2012). Therefore, the quantification of the impacts of irrigation on land-atmospheric fluxes will improve our knowledge on how human disturbances impact the natural water, energy and carbon cycles, but will also contribute to preventing the potential destruction of the environment and resources caused by excessive water irrigation.

However, since the effects of irrigation differ depending on the location, and the crop response to irrigation varies during distinct growth phases, it is not an easy to thoroughly evaluate the effects of irrigation on land-atmosphere fluxes. Several investigations confirmed the repercussions of irrigation on land-atmosphere fluxes (Lobell et al., 2008; Leng et al., 2013; Cook et al., 2011; 2015; Puma and Cook 2010). Boucher et al. (2004) conducted simulations based on the General Circulation Model (GCM) to quantify the effects of irrigation on the atmospheric water vapor content on a global scale. They revealed the water vapor flux from irrigation could reach as high as 300 kg $m^{-2}$ year$^{-1}$ in eastern China and northern India,

where irrigation was most intensive. Sacks et al. (2008) conducted a land-atmosphere coupling simulation using the Community Atmosphere Model (CAM) and the Community Land Model (CLM) to study the global impact of irrigation on sensible heat flux and near-surface temperature. They demonstrated that irrigation influenced the vapor flux and also reduced the sensible heat flux over the northern mid-latitude regions (e.g. the central United States, southeast China and portions of

southern and southeast Asia); thus cooling the global average temperature by ~0.5 ℃. These findings were corroborated by Pokhrel et al. (2012) using the Minimal Advanced Treatments of Surface Interaction and Runoff (MATSIRO) land model combined with the Gravity Recovery and Climate Experiment (GRACE) satellite observations. The authors demonstrated that irrigation produced a maximum increase of 50 W m$^{-2}$ in latent heat flux averaged over the months of June and August. Leng et al. (2014; 2015) focused on the groundwater-fed irrigation and incorporated a groundwater withdrawal scheme into

the CLM4. The authors calibrated and run the coupled model over United States and globe, and underscored the importance of local hydrologic feedbacks in governing hydrologic response to anthropogenic change.

    The global investigations described above can define the critical zones affected by irrigation; however, some shortcomings are apparent. First, common methods of large-scale modeling use soil moisture or crop ET deficiencies to estimate the temporal and spatial distributions of irrigation. However, in reality, the determination of the irrigation location is much more

complicated and locally dependent. The water allocation is related to the water demand and available water, but also closely involves the local water policy and residents' habits and customs. Secondly, in reality, irrigation does not occur in one spatially contiguous layer of moisture but rather as a patchwork of individual fields covering hundreds of hectares. The large-scale modeling often displays a resolution of 0.1 ° to 1.0 ° (10 km to 100 km at the Equator), which is too large to describe the irrigation in a patchwork distribution. Several recent studies have focused on the consequences of irrigation on a

regional scale, such as in South Asia (Shukla et al., 2012), or in the Northern China Plain (Zou et al., 2014, 2015). However, the model grid size (2 ° and 25 km) is still unsatisfactory to capture the feature of each irrigated patchwork. Zeng et al. (2016) incorporated the schemes describing groundwater lateral flow and human water regulation into land model CLM4.5 and conducted high-resolution (about 1-km) simulations to quantify the effects of human water-related activities on land processes. It showed impressive results that land model could be used in high-resolution and basin scale modeling. However,

the prognostic carbon-nitrogen cycle and crop model in CLM4.5 were not active in Zeng et al. (2016), which make its results with high-level uncertainty and excluded from impacts of irrigation on carbon cycle. To overcome these difficulties above and go further steps, our study focused on a typical semi-arid irrigated basin; the Heihe River Basin situated in northwestern China, and conducted a high resolution simulation (1 km) using the state-of-art CLM4.5 land model with active carbon-nitrogen cycle and crop model, and incorporating a high-quality irrigation dataset to quantify the impacts of irrigation

on the LH, SH and NEE parameters from land to atmosphere. The objectives of our study are: (1) testing the model performance of CLM4.5 land model under the active carbon-nitrogen cycle, crop model, and high-quality irrigation input data, (2) quantify the effects of irrigation on the LH, SH and NEE parameters and, (3) decipher the relationship between the irrigation and the changing fluxes. Besides the scientific objectives above, our work is a further step in producing a high-resolution land surface simulation model that represents crop growing at a basin-scale. The model can bring valuable

support to future investigations aiming to improve the representation of hydrological and ecological processes in Earth system models (Clark et al., 2015a, 2015b, 2015c; Fan, 2015).

Background information related to the domain of study is given in Sect. 2. The model is introduced in Sect. 3, including the data sources for irrigation and the experimental design. Sect. 4 presents the results of the simulations, while the discussion and conclusions are given in Sect. 5.

## 2 Domain of Study

The Heihe River Basin is the second largest inland river basin located in China (Figure 1). The basin spans an area from 96°42' E to 102°00' E long. and from 37°41' N to 42°42' N lat. (Lu et al., 2003). The area covered by the basin is about 116,000 km$^3$ and lies east of the Shule River and west of the Shiyan River basins (Chen et al., 2005). The Heihe River Basin upstream and middle reaches sit in a semi-arid area, whereas the downstream reaches retain an arid climate all year round.

The Heihe River Basin is characterized by different geomorphologic features. From south (upstream) to north (downstream) we observed the Qilian Mountains, Hexi Corridor and Alxa High-plain. The climatic condition and water resource distribution also differ. In the upstream basin where mountainous areas predominate, the average precipitation is ~200 mm year$^{-1}$ at elevations ranging from 2,000 m to 3,000 m, and ~500 mm year$^{-1}$ from 3,000 m to 5,500 m. The high-altitude region is the main water resource of the entire basin (Wu et al., 2010). In the middle reaches of the basin, where the elevation shows a North-South decrease from 2,000 m to 1,000 m, the precipitations diminish correspondingly from 200 mm year$^{-1}$ to less than 100 mm year$^{-1}$ (Li et al., 2001). The middle reaches contain sufficient light-heat resources and is the major area of crop cultivation. Corn is the main cultivated vegetable occupying 62% of the arable land. Barley, wheat, and cotton are also important crop types of the Heihe River Basin. The scarce precipitation forces the intense extraction of irrigation water from the mainstream of the Heihe River. The exploitation of groundwater in the middle reaches also increased during recent years to reduce the removal of surface water and maintain a base flow for the downstream environment, in concordance with the water policy of the "water diversion scheme" set by the local government in 1997. The downstream area of the basin is formed by the Alxa High plain at an altitude of about 1,000 m. The plain is extremely arid characterized by only 42 mm year$^{-1}$ of annual precipitation based on data from local meteorological stations (Qi and Luo, 2005).

## 3 Model Description and Experimental Design

### 3.1 Community Land Model 4.5

Generally The CLM4.5 land model used in this study was developed by the National Center for Atmospheric Research, USA (Lawrence et al., 2011; Oleson et al., 2013). The model constitutes the land component of the Community Earth System Model (CESM) 1.2.0 (Gent et al., 2011; Hurrell et al., 2013). The CLM4.5 simulates the exchange of radiation, momentum,

heat and water vapor flux between the land and the atmosphere. It also models hydrologic processes (including precipitation interception, soil infiltration, runoff production, soil water movement, aquifer recharge and snow dynamics); vertical energy transfer within soil and snow; and other bio-geophysical processes (Lindsay et al., 2014). Important bio-geochemical processes, which include the carbon-nitrogen cycles, vegetation photosynthesis and respiration, phenology, decomposition, 5 and fire parameterization (among others), are also explicitly defined by the CLM4.5 model.

Additionally, the CLM4.5 incorporated an interactive crop management model based on the Agro-IBIS crop module. The latter simulates the crop growth and its effects on land processes. The model sets specific parameters for corn, temperate cereals and soybean and includes a crop phenology algorithm for different growth phases of planting, leaf emergence, grain fill and harvest. The carbon-nitrogen distribution in each crop tissue (leaf, stem, root and grain) is also different to that of 10 natural vegetation. The crop module in CLM4.5 has been thoroughly tested by Levis et al. (2012) and was found to be highly performant for simulating the carbon fluxes from land to atmosphere.

CLM4.5 is renowned for its ability to model large-scale events, especially in reproducing land-atmosphere fluxes. However, few studies to date tested the model on smaller-scale cases with a high resolution rather than 1-km. In theory, since the bio-geophysical and bio-geochemical aspects of the model are relatively complex, CLM4.5 will be suitable for 15 small-scale events if the input data is comprehensive enough. This study thus benefits from the complexity of the model and the numerous high resolution input data available for the Heihe River Basin to demonstrate the adequacy of the land model at a small scale.

In this study, we adopted a modified version of CLM4.5 which was improved by Zeng et al. (2016). This version of CLM4.5 was coupled with a groundwater lateral flow scheme which enable the model to explicitly capture the water table 20 pattern following terrain-driven lateral flow. A scheme describing human water regulation was also included in the model. More detail information about the improved CLM4.5 can be found in Zeng et al. (2016).

**3.2 Brief introduction for the irrigation module of the improved CLM4.5 model**

To investigate the irrigation effects on land-atmosphere fluxes, the irrigation scheme in the land model is a key part. The original version of CLM4.5 includes the option to irrigate croplands. However, this version is not appropriate for our 25 fine-scale study because: (1) the resolution of the original irrigation database in CLM4.5 is 15 arc min (about 0.083 ° at the Equator) which is too coarse and not precise enough for our high-resolution (1 km or 0.0083 ° at the Equator) modeling, (2) the irrigation module does not include groundwater extraction which could impact the land-atmosphere fluxes by modifying the water table level and soil moisture content (Di et al., 2011; Xie et al., 2012), (3) the irrigation water included in CLM4.5 is removed from local runoff sources rather than from nearby rivers. This procedure does not reflect the real water diversion 30 process, especially when model resolution is fine enough to distinguish the river grid cells from others. Although some key works conducted by Leng et al. (2014; 2015) have extended the CLM's ability in modeling anthropogenic groundwater exploitation, the officially public version of CLM4.5 has not included such an important human activity at this time.

The improved CLM4.5 developed by Zeng et al. (2016; 2017) incorporated an irrigation scheme which could settle the

aforementioned points. The diagram of the complete module is presented in Figure 2. In the model, the process of groundwater withdrawal for irrigation is described as:

$$\begin{cases} d' = d + \dfrac{Q_g \times \Delta t}{s} \\ W' = W - Q_g \times \Delta t \end{cases}, \tag{1}$$

in which $\Delta t$ (T) is the time step of the model; $s$ is the aquifer specific yield provided by CLM4.5; $Q_g$ (L/T) is the groundwater pumping rate, and $d$ (L) and $d'$ (L) are, respectively, the simulated groundwater table depth before and after accounting for anthropogenic groundwater exploitation, and $W$ (L) and $W'$ (L) are, respectively, the simulated aquifer water storage before and after accounting for anthropogenic groundwater exploitation. The surface water diversion for irrigation in the model is described as:

$$S' = S - Q_s \times \Delta t , \tag{2}$$

in which $Q_s$ (L/T) is the anthropogenic surface water intake, and $S$ (L) and $S'$ (L) are, respectively, the original surface water stored in a river (as calculated by CLM4.5 coupled with the River Transport Model) and the updated value after subtracting the anthropogenic demand. If the local surface water intake $Q_s \Delta t$ is greater than local surface water storage $S$, the deficit is satisfied by extracting surface water from nearby grid cells. The water abstracted from aquifer and river is applied directly to the ground surface, bypassing the canopy interception as:

$$Q_i' = Q_i + Q_g + Q_s , \tag{3}$$

in which $Q_i$ (L) and $Q_i'$ (L) are, respectively, the simulated water input into the soil surface before and after accounting for anthropogenic water application. The irrigation water losses to runoff and aquifer would be calculated automatically by CLM4.5 with the updated water input to soil $Q_i'$ which combines both the precipitation and irrigation. The $Q_g$ and $Q_s$ are determined by the external high-quality irrigation dataset which will be described in Section 3.3.

The irrigation water would moisten the soil and operate on crops, alleviating the water stress occurring in semi-arid areas and improving the conditions of growth. This in turn would subsequently modify the water, energy and carbon fluxes of the land and the atmosphere. The irrigated water is distributed uniformly throughout the day, reproducing the process of flooding irrigation. It should be noticed that the water balanced is closed in the model because the total withdrawal amount of the water (from surface and underground) is equal to the whole consumption in irrigation.

### 3.3 Irrigation database for the Heihe River Basin

One advantage of basin-scale modeling is the application of precise irrigation data provided by local water departments to reduce the uncertainties from input data. We applied the Monthly Irrigation Datasets (MID) provided by the Cold and Arid Regions Science Data Center (CARSDC) at Lanzhou to the model input. The MID is a high-resolution (30' or 1km) monthly

gridded dataset collected from 1981 to 2013.The database differentiates the irrigation water provided by surface water and groundwater. The MID was assembled by collecting a large number of irrigation statistics from local administrations, historical literature and farmland surveys at the province level, and by downscaling the collected data to 1 km grid cells using the retrieved ET from the remote sensing dataset ETWatch (Wu et al., 2012). Other data sources were referenced when

the dataset was created. The MID has been verified successfully by comparison with independent databases provided by the Water Resource Bullet of the Gansu Province and from statistics contributed by several irrigation districts. Figure 3 shows the MID spatial distribution of the irrigation rate occurring during the four seasons. The irrigation is principally applied on farmland associated with the middle reaches of Heihe River Basin, with summer being the season with the most intensive irrigation. There is almost no irrigation occurring during winter.

Since the irrigation rate was prescribed before the simulations, it does not interact with the crop water stress (reflected by soil moisture or crop transpiration) in the model. This is one of the disadvantages of our simulation because in reality the irrigation rate is low when soil water is rich. Arguably, the irrigation rate from MID was based on the actual evapotranspiration and historical irrigation records, so in fact it accounted for the water stress in an implicit way. Moreover, on local level, besides the crop water stress, the water supply from upstream, water policy and residents' habits and customs

also play key roles in the determination of irrigation rate, therefore it is reasonable to apply an external high-quality dataset to quantify irrigation in the high-resolution simulations.

### 3.4 Experimental design

Two simulations were conducted using the improved CLM4.5 (Zeng et al., 2016) on the Heihe River Basin. The first simulation named "CTL" forms the control simulation assuming no irrigation is applied, and the second one named "IRR" is

the simulation including the effects of the irrigation dataset referred to in section 3.3. The simulation period was set from 1981 to 2013, corresponding to a period during which the input irrigation dataset was collected, with a time step of 1,800 seconds. The simulations spatial resolution was fixed at 30 arc-seconds (0.0083 °). To satisfy the high-resolution process, we replaced the CLM4.5 land cover data with the Multi-source Integrated Chinese Land Cover map available at a 1 km resolution (Ran et al., 2012). We used the Heihe Watershed Allied Telemetry Experimental Research (HiWATER) Land cover

map having a 30m resolution (Li et al., 2013; Zhong et al., 2014) to identify the specific crop types (corn, cereals, soybean or others) from each crop grid. The soil dataset (content of clay, silt, sand and organic matter) included in the CLM4.5 was replaced by the China Soil Characteristics Dataset with a 1 km resolution (Shangguan et al., 2012). The databases were provided by CARSDC (http://westdc.westgis.ac.cn/). It should be noticed that in the high-resolution surface dataset, only one kind of landscape (e.g. bare soil, different plant function types, lake or urban) existed within a model grid cell, indicating that

most of the sub-grid structure were not applied. To obtain the impacts of irrigation on the crop carbon fluxes, the fully prognostic carbon and nitrogen cycles modules accompanied by the interactive crop management module in CLM4.5 were activated. The atmospheric forcing dataset was acquired from the Data Assimilation and Modeling Center for Tibetan Multi-spheres, Institute of Tibetan Plateau Research (ITP) of the Chinese Academy of Sciences (Yang et al., 2010), with a

spatial resolution of 0.1 ° and a temporal resolution of 3 h. Before beginning the proper simulations, the CLM was run for a 700-year period (also using the ITP atmospheric forcing dataset, but in cycle, without irrigation) to produce model variables attaining their natural equilibrium state (especially for each carbon and nitrogen pool) following the CLM user guide (Kluzek, 2013). The CTL and IRR simulations are then computed using the initial files generated by the 700-year spin-up.

## 4 Results

### 4.1 Validation

The performance of the implemented model has been tested in Zeng et al. (2016) by multiple comparisons with observation from the Eddy Covariance (EC) and Automatic Weather (AW) systems, groundwater wells and remote sensing. In this study, we additionally compared our simulated results with National Aeronautics and Space Administration's Gravity Recovery and Climate Experiment (GRACE) satellites data (Tapley et al. 2004), retrieved ET and Net Primary Productivity (NPP) acquired from remote sensing.

Figure 4 shows the basin-averaged terrestrial water storage (TWS) from GRACE and the modeled TWS from CTL and IRR in the period of 2002-2013 when GRACE data were available. From the picture, both CTL and IRR captured the general variations of TWS showed by GRACE. However, their linear trends were different. The result from CTL shows a minor regression coefficient of -0.10 cm year$^{-1}$, while it shows -0.19 cm year$^{-1}$ in the IRR, much closer to the -0.20 cm year$^{-1}$ displayed by the GRACE data. This comparison demonstrated that irrigation from groundwater exploitation has significantly depleted the terrestrial water storage in the basin, and this effect could be reproduced when groundwater irrigation was included in the model.

Then we compared our modeled streamflow with the observations. Although the interested variables in this study are the energy and carbon fluxes, the valid streamflow can indicate the model takes right amount of water from the river for irrigation. The gauged discharge was obtained from the Yingluoxia hydrologic station located in Yingluoxia Gorge, the inlet of midstream of Heihe River (shown in Figure 1). The discharge data were provided by Liu et al. (2016). Figure 5 shows the time series for the modeled and observed daily streamflow over Yingluoxia Gorge from 2002 to 2004. It shows that our model can generally truly reproduce the magnitude and seasonal variability of the discharge, and the correlation coefficient is about 0.7. However, the error between the simulation and observation was obvious. It indicates that the River Transport Model of CLM4.5 could be substantially improved and the reservoirs should be included. Taking into consideration that the subjects in the study are the land-atmosphere fluxes rather than streamflow and the river only plays the role of water source for irrigation in the simulation, the modeling skill at this time could be accepted.

We then compared the modeled ET with the remote sensing dataset from ETWatch (Wu et al., 2012). Although Zeng et al. (2016) has done similar work, this comparison is different for that the modules of prognostic carbon-nitrogen cycles and interactive crop management in CLM4.5 was active and the input irrigation data were much more realistic than it used in Zeng et al. (2016). The ETWatch retrieves actual ET using multi-sources remote sensing data and multiple inversion

algorithms such as TSEB (Norman et al., 1995; Anderson et al., 1997), SEBS (Su, 2002) and SEBAL (Bastiaanssen et al., 2005). It has been independently verified using various procedures on different cultivated fields and landscapes. Figure 6 shows the climatologic ET distribution obtained from 2000 to 2003 by ETWatch, CTL and IRR over the Heihe River Basin. The period was selected based on the duration of the ETWatch. Figure 6a features a contiguous area with a high level ET

obtained by ETWatch along the middle reaches of Heihe River. The high level ET results from intensive irrigation (see Figure 3). This pattern did not appear in the CTL results (Figure 6b). However, it was reflected in the IRR results when irrigation was included (Figure 6c). This again demonstrates the importance of incorporating irrigation in the studies of the land-atmosphere water vapor flux (latent heat flux). South of the Heihe River Basin, in the upstream reaches where arctic grass grows without irrigation, both the CTL and IRR poorly captured the ETWatch ET pattern, indicating the CLM model

needs improvement for carbon-water cycle simulations in cold regions.

We then compared our NPP results derived from remote sensing data. The NPP is the difference between the Gross Primary Productivity (GPP) and the autotrophic respiration of plants. It almost equals the negative NEE, but excludes the heterotrophic respiration. Lu et al. (2009) build-up the applied NPP dataset, using the high-resolution remote sensing SPOT-VEGTATION NDVI satellite data and a Monteith-type parametric NPP model. The annual database spans the period

of 1998 to 2002. Figure 7 illustrates the climatologic NPP distribution over Heihe River Basin obtained from remote sensing, CTL and IRR during 1998 to 2002. The high-level NPP occurring in the remote sensing imagery over the middle reaches (Figure 7a), corresponded to irrigated croplands as shown in Figure 3. Furthermore, the retrieved arctic grass NPP in the upstream region was also relatively high. Figure 7b, presenting the CTL results, displays a very low NPP (less than 200 gC $m^{-2}$ $year^{-1}$) across the entire Heihe River Basin. However, when intense irrigation took place, a high-level NPP emerges in

the IRR model over the middle stream (Figure 7c), consistent with the remote sensing dataset. The comparison demonstrates the model ability to simulate NEE over irrigated areas and to establish the significant effects of irrigation on NEE. The CTL and IRR models could not reproduce the high-level NPP in the upstream region, indicating the CLM model needs further modifications to improve its predictions for arctic regions.

### 4.2 The effects of irrigation on land-atmosphere fluxes

Section 4.2 validated the CLM4.5 model when a well-developed irrigation module is combined with a high-quality irrigation dataset to reproduce the land-atmosphere interactions. We then studied the effects of irrigation on the LH, SH and NEE utilizing high-resolution results obtained from the IRR and CTL runs. Figure 8 displays the differences between IRR and CTL climatic spatial patterns of the LH, SH and NEE during the spring, summer and autumn months, averaged for the 1981–2013 period. Figure 8a to 8c demonstrate that the effects irrigation produced on LH corresponds to an enhanced LH

(or ET) occurring over most of the irrigated areas. During summer, the response to irrigation was prominent over a vast zone of the middle reaches, with the LH locally increasing by ~100 W $m^{-2}$ (or ~3.5 mm $day^{-1}$ for ET). The consequences during spring and autumn were less important, producing a LH increase of 10–30 W $m^{-2}$ and 20–40 W $m^{-2}$, respectively over the middle reaches. The LH intensification is attributed to increased evaporation from a wetted soil surface and enhanced ET

from crop submitted to less water stress. The LH absorbing more energy from land, the GT is cooled, and the SH from land to atmosphere is reduced correspondingly. The process is shown in Figure 8d to 8f. The SH is lessened during the summer over a large irrigated area of the midstream Heihe Basin in conjunction with LH. The reduction reaches ~60 W m$^{-2}$. The SH reduction is about 30 to 60 W m$^{-2}$ during the spring, with a very weak effect during autumn (less than 20 W m$^{-2}$). Figure 8g to 7i exhibit the differences in NEE for three seasons. During autumn, when the crop photosynthesis is weak, there is little difference between the IRR and CTL model results (Figure 8i). However, the effects of irrigation on NEE diverged significantly during the summer and spring months. This is caused by that less water stress induced by irrigation enhancing the GPP and crop respiration. In the growth phase of planting and leaf emergence during the spring months, the crop leaves were not fully grown and generated a relatively low rate of photosynthesis. The increase in respiration caused by irrigation was higher to that of GPP, leading to a positive change of NEE (e.g. 0.4–0.8 gC m$^{-2}$ day$^{-1}$; the positive direction indicating a transfer of carbon from land to atmosphere) over most of the middle irrigated reaches of the Heihe Basin. During summer, when the crop shifts from the leaf emergence phase to the grain fill phase, the Leaf Area Index (LAI) and photosynthesis rate are at their maximum for the year. Therefore, after irrigation was included in the models, the GPP increase could become higher to that of respiration, generating a negative change of NEE (less than -0.8 gC m$^{-2}$ day$^{-1}$, the negative change indicates the land ecosystems fixed more carbon from atmospheric $CO_2$).

Scatter plots, displayed in Figure 9, show the differences of fluxes (LH, SH and NEE) between the IRR and CTL models against the irrigation rate during spring, summer and autumn. The fluxes and irrigation rates were averaged over the 1981–2013 period; each dot representing an irrigated model grid cell of the Heihe River Basin. Figure 9a-9c present the relationship between ΔLH (LH between IRR and CTL) and irrigation rate. During the spring, when the irrigation rate is less than 2 mm day$^{-1}$, the ΔLH values increase rapidly with the augmentation of irrigated water usage. However, when the irrigation rate is greater than 2 mm day$^{-1}$, the ΔLH increase disappears. This is related to the water condition meeting the need of potential ET, when excessive irrigation would no longer produce an ET increase. A similar process occurred during summer and autumn (Figure 9b and 9c), but the irrigation rate thresholds increased from 2 mm day$^{-1}$ to ~4 mm day$^{-1}$. Figure 9d-9f display decreasing ΔSH (SH between IRR and CTL) values correlated with an increase of the irrigation rate which are similar for the three seasons. However, ΔSH remains constant (~ –60 W m$^{-2}$ for spring, ~ –80 W m$^{-2}$ for summer and ~ –20 W m$^{-2}$ for autumn) when the irrigation rate is greater than ~5 mm day$^{-1}$. Figure 9g-9i also present a positive correlation between the ΔNEE (NEE between IRR and CTL) values and irrigation rate during the spring, with the irrigation rate increasing from 0 to 5 mm day$^{-1}$, corresponding to ΔNEE values ranging from 0 to 1.5 gC m$^{-2}$ day$^{-1}$; once again demonstrating the stronger effects of irrigation on crop respiration during spring relative to GPP. In contrast, during summer (Figure 9h), when the irrigation rate goes from 0 to 5 mm day$^{-1}$, the ΔNEE varies from 0 to –1.5 gC $^{-2}$ day$^{-1}$, indicating that the effect of irrigation on GPP is stronger than it on respiration so that the ΔNEE is negatively affected (more carbon is fixed by crop) during the summer. There is no clear correlation between ΔNEE and the irrigation rate during autumn (Figure 9i). The ΔNEE values can augment or diminish when the irrigation usage is intensified.

One of the important findings above is the threshold of 5 mm day$^{-1}$ irrigation rate, above which the effects of irrigation on

LH and SH do not change considerably. This is tightly related to the semi-arid climate of the basin. Over the middle reaches where typical continental climate dominates, precipitation is scarce and temperature is high during the growing season. The most controlling factor for actual ET is the water availability. Therefore, when the water demand for potential ET (PET) is being satisfied with irrigation, the actual ET and LH would rapidly increase and the SH would correspondingly decrease. When the irrigation rate exceeds a certain threshold, ~5 mm day$^{-1}$ in this study, the water demand of PET is fully satisfied and the controlling factor of actual ET converts from water availability to energy availability. Thus the LH and SH are no more sensitive to the increased water supply when irrigation rate exceeds ~5 mm day$^{-1}$. It also explains why the irrigation threshold in summer is higher than it is in spring and autumn (shown in Figure 9): The water demand needed to elevate the actual ET to the PET is highest in summer due to the hot weather. To demonstrate the analysis above, a scatter plot (Figure 10) showing the differences between the PET and actual ET (PET-ET) against the irrigation rate in summertime was displayed. The ET was obtained from the IRR simulation, and the PET was estimated from the near surface temperature (also produced by the IRR) via the method of Thornthwaite (Thornthwaite, 1948). The Thornthwaite method was chosen here because the only input variable it needs to calculate the PET is the near surface temperature, a highly credible variable reproduced by the land surface model. A systematic error (about -1.5 mm day$^{-1}$) in the Thornthwaite method (Hashemi and Habibian, 1979; Pereira and Camargo, 1989) has been removed in the Figure 10 to not allow the PET-ET being negative. The Figure 10 shows that when irrigation rate less than ~5 mm day$^{-1}$, the PET-ET decreases with irrigation increases, and when the irrigation rate exceeds ~5 mm day$^{-1}$, the PET-ET tends to be zero, demonstrating that the water demand of PET can be satisfied with an irrigation rate of ~5 mm day$^{-1}$.

We then investigated the inter-annual and seasonal variability on land-atmosphere fluxes caused by irrigation. Figure 11a-11e show a time evolution of the annual irrigation rate, precipitation, Crop Temperature (CT), and some land-atmosphere fluxes obtained from 1981 to 2013. The data was averaged over all crop grid cells. Figure 11a illustrates a slightly reduced annual irrigation rate from 1.8 to 1.6 mm day$^{-1}$ from 1981 to 2013, whereas the precipitation rate increased moderately from 0.3 to 0.5 mm day$^{-1}$. The total water input (irrigation plus precipitation) to the cropland was almost constant over the last 30 years. Figure 11b reveals a significant increasing trend of annual CT directly attributed to global warming, whereas Figure 11c illustrates an important augmentation in annual LH during 1981–2013 that could not be explained by steady irrigation or precipitation trends (Figure 11a). In contrast, the increasing LH trend could be linked to the growing CT (shown in Figure 11b). Therefore, the inter-annual LH trend was influenced more by the temperature (radiation limitation) than the irrigation rate (soil moisture limitation). In Figure 11d, the SH time evolution plot also displays a significant decreasing trend linked to the temperature augmentation. Figure 11e presents the negative consequence of irrigation on the annual NEE (e.g. more carbon is transferred from atmosphere to land). Furthermore, the NEE increases during the IRR modeling and decreases in the CTL modeling, resulting in a ΔNEE augmentation (IRR-CTL) from 1981 to 2013. In 2010, the NEE in IRR even became higher to that in the CTL. Figure 11f-10j present the seasonal variations in annual irrigation, precipitation, CT, and land-atmosphere fluxes. The data obtained from 1981 to 2013 was averaged over all crop grid cells in the Heihe River Basin. The LH and SH response to irrigation is substantial in the growth season occurring from April to

October (Figure 11h and 10i), whereas it is negative for NEE from May to September, and positive in March, April and November (Figure 11j)

Figure 12a shows the temporal correlation coefficients ($R$) obtained for the irrigation rate versus the differences in land-atmosphere fluxes for spring, summer and autumn ($\Delta$LH, -$\Delta$SH and $-\Delta$NEE, the negative signs before $\Delta$SH and $\Delta$NEE were applied to make $R$ positive). The correlations between the irrigation rate and turbulent fluxes variations were very poor and did not pass the Student's t-test at a 95% confidence level. This again proves irrigation significantly modified the land-atmosphere fluxes for each year, but the inter-annual variability did not have much impact on the fluxes. However, for the spring NEE, the obtained $R$ value was 0.6 and passed the Student's t-test, suggesting more irrigation water applied during the spring can cause more carbon absorption by the crop. It is attributed to an earlier associated leaf emergence phase to better water conditions for the crop. The correlation coefficient ($R\sim0.4$) between the CT (from IRR) and the differences in fluxes passed the significant test and is reproduced in Figure 12b which shows the $\Delta$LH and $\Delta$SH values are associated with CT during the spring period. Figures 9 and 10 demonstrate the effects of irrigation on the land-atmosphere systems are not solely linked to irrigation amount; other factors (such as the global warming) also play important roles that cannot be ignored.

## 5 Conclusions and Discussion

In this study, we performed two modeling experiments (CTL and IRR) over Heihe River Basin of China conducted by improved CLM4.5 with high-quality irrigation data collected from 1981 to 2013 to quantify the effects of irrigation on land-atmosphere fluxes. Comparing the models' results to fluxes measurements obtained from remote sensing, demonstrated their reliability. The consequences of irrigation on LH, SH and NEE were analyzed based on the simulations results and we studied the relation between the fluxes and the intensity of irrigation.

The principal conclusions brought by our investigations are: (1) The effects of irrigation on the LH and SH parameters are strongest during summer with an increase in LH of $\sim$100 W m$^{-2}$ and a decrease in SH of$\sim$60 W m$^{-2}$ over intensely irrigated areas. However, the effects of irrigation are less important during spring and autumn because of a lesser application of irrigated water, (2) When the irrigation rate is less than 5 mm day$^{-1}$, the LH generally increases and the SH is reduced with an increasing irrigation rate. When the irrigation rate exceeds the threshold of 5 mm day$^{-1}$, its effect on LH and SH is subdued, (3) During spring, irrigation produces more carbon transferred from the land to the atmosphere, increasing NEE by 0.4–0.8 gC m$^{-2}$ day$^{-1}$, whereas during summer, irrigation facilitates the crop fixation of carbon from atmospheric $CO_2$, decreasing NEE by $\sim$0.8 gC m$^{-2}$ day$^{-1}$ and, (4) The results of irrigation on land-atmosphere fluxes are also linked to other parameters (especially the temperature) that also play important roles on LH, SH and NEE.

The depletion of terrestrial water storage in the basin needs to be stressed at here. From Figure 4, both the simulation IRR and GRACE satellite reflect a water loss of about 2 mm per year. This depletion is critical to the inland basin where mountainous glacial water and groundwater supply play essential roles in irrigation. To deal with the water depletion, it is

necessary to determine balances between economic development and water exploitation, and between surface water diversion and groundwater pumping. Wu et al. (2016) has set up some meaningful studies for optimizing conjunctive use of surface water and groundwater for irrigation to address human-nature water conflicts. More researches should focus on the future water resources problem in the basin.

5    Our study demonstrates that irrigation plays the essential role on the exchange of water, energy and carbon from the land to the atmosphere. Limitations and assumptions inherent to the CLM4.5 model and atmospheric forcing data can bring some uncertainties to our results (Bonan et al., 2011, 2013; Mao et al., 2012; Wang et al., 2013). Our module currently cannot distinguish among sprinkler irrigation, flooding irrigation and drip irrigation, while in reality there should be different treat ways for different crop types. This may lead to a slight over-estimation of water reaching the topsoil. Moreover, currently 10 there is no scheme for the reservoir in the basin for irrigation purpose, and the groundwater pumping is assumed executed on the location where it is consumed. These simple and unrealistic assumptions will certainly take some uncertainties into our results and show the future directions for model development. However, our results provide a high level of confidence because: (1) CLM4.5 is one of the world's most advanced and widely used land model verified especially for its ability to reproduce land-atmosphere fluxes, (2) The simulations resolution is high enough (1 km) to show patchworks of irrigated 15 areas, and the input irrigation database is comprehensible and reliable and, (3) There is consistency between the simulated models output and the observation of remote sensing. Another limit is that the high-quality irrigation data is available for this particular catchment, but not at continental scale. This limits the applicability of this experiment to larger domains before we produce high-resolution irrigation dataset over a larger scale.

    Future studies should focus on three aspects. First, other important land-atmosphere fluxes, such as the momentum, 20 nitrogen and oxygen fluxes are also related to irrigation and deserved to be investigated further. Secondly, irrigation may modify the climate affecting the natural processes and anthropogenic activities on land. This land-atmosphere interaction needs to be explored using regional or global climate models. Third, this study conducted a high-resolution modeling over a typical irrigated basin. The ultimate goal is to perform high-resolution modeling on a continent or even global scale using the advanced power of supercomputers.

25 **Code availability**

The model code is available upon request. Please contact Zhenghui Xie at zxie@lasg.iap.ac.cn.

**Acknowledgements** This work was jointly funded by the Key Research Program of Frontier Sciences, Chinese Academy of Sciences (grant Nos. QYZDY-SSW-DQC012) and by the National Natural Science Foundation of China (grant Nos. 30 41575096). The irrigation dataset was provided by the Cold and Arid Regions Science Data Center at Lanzhou (http://www.heihedata.org/data/66322216-b99a-4c54-a130-a0c3ef0ed972). The atmospheric forcing data was obtained through the Data Assimilation and Modeling Center for Tibetan Multi-spheres, Institute of Tibetan Plateau Research,

Chinese Academy of Sciences (http://westdc.westgis.ac.cn/data/7a35329c-c53f-4267-aa07-e0037d913a21). The databases from the ET and NPP remote sensing data of were provided by the Cold and Arid Regions Science Data Center at Lanzhou, China (http://westdc.westgis.ac.cn). We would like to thank Xing Yuan and Xiangjun Tian for their assistance with this work and their helpful discussions. We also thank Guoyong Leng and the other anonymous reviewer for the helpful comments that improved the manuscript.

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

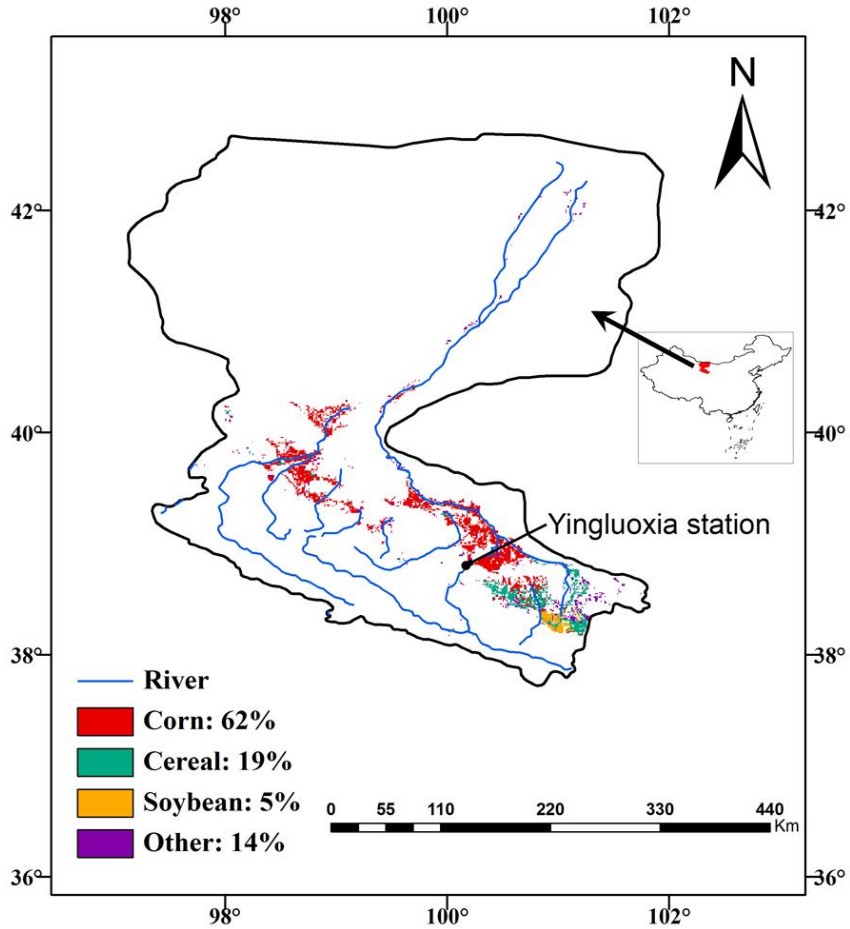

**Figure 1: Study area and location of the Heihe River Basin in northwestern China, the spatial pattern of crops in the basin, and the location of Yingluoxia hydrologic station.**

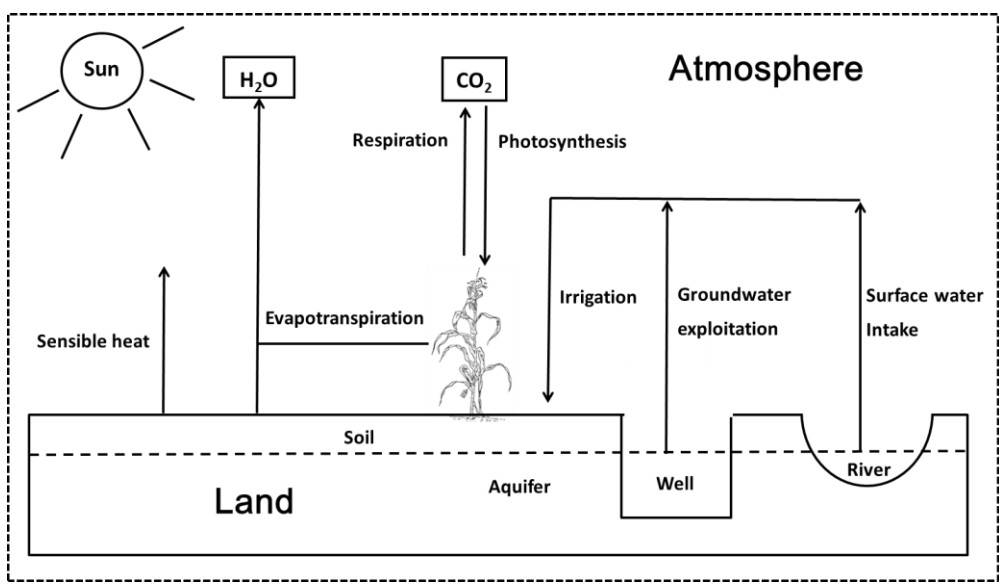

**Figure 2: Diagram of the irrigation module showing how irrigated water impacts on land-atmosphere fluxes.**

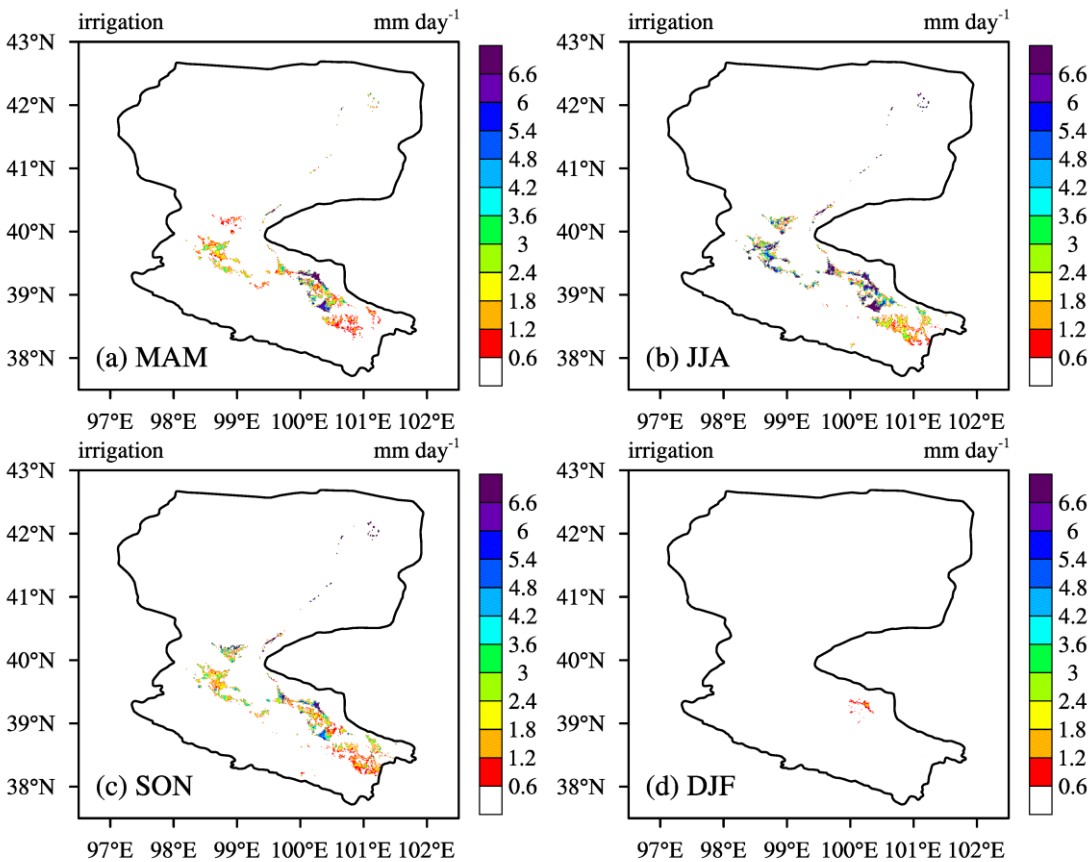

**Figure 3: Climatologic spatial distribution of the irrigation rate from Monthly Irrigation Datasets in (a) spring, (b) summer, (c) autumn and, (d) winter from 1981 to 2013.**

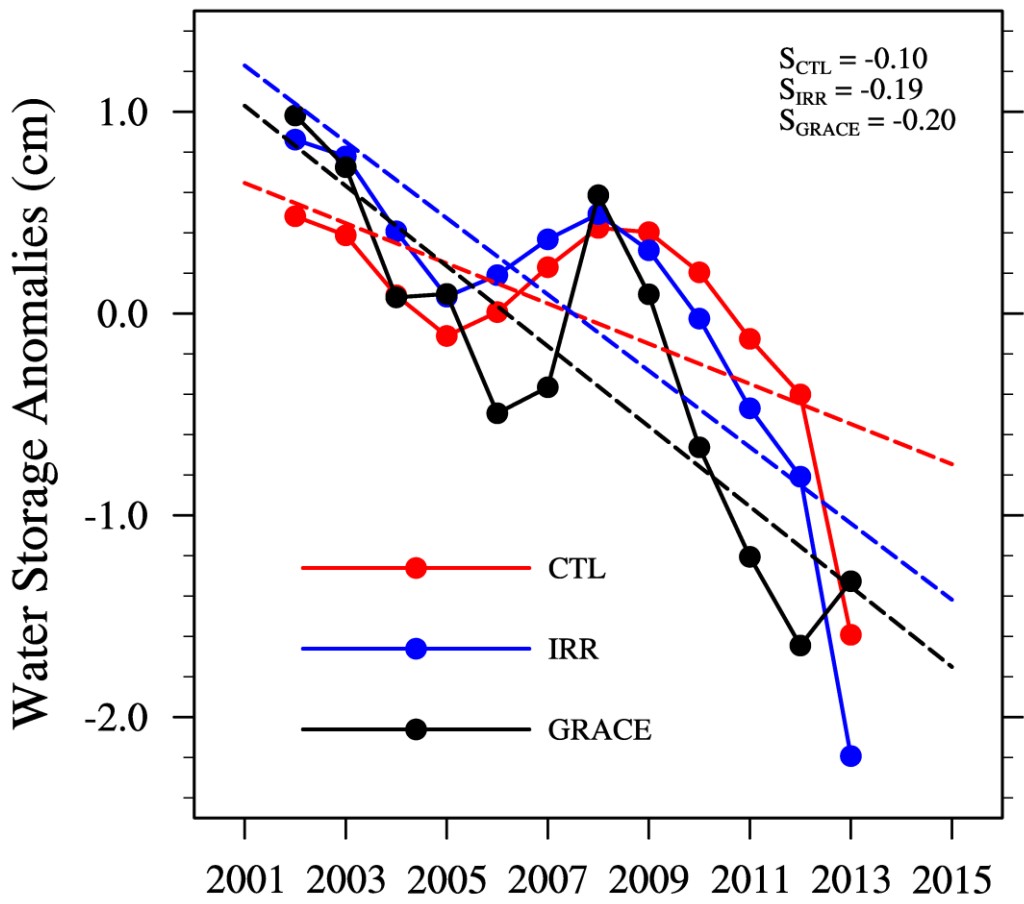

**Figure 4: Basin-averaged terrestrial water storage (TWS) from GRACE and the modeled TWS from CTL and IRR in the period of 2002-2013.**

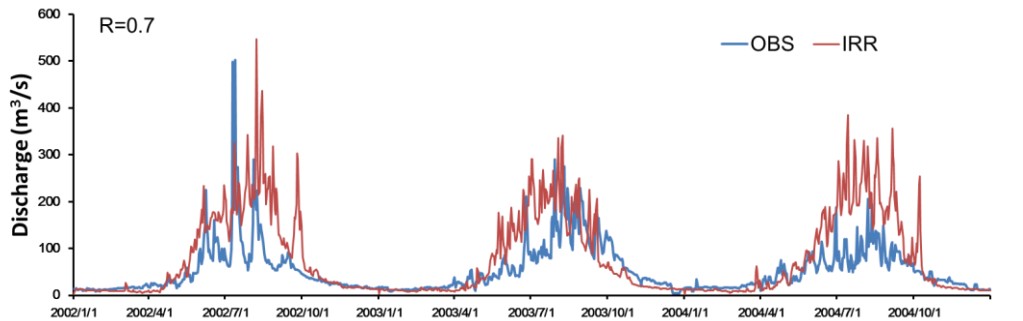

**Figure 5: Time series for the modeled and observed daily streamflow over Yingluoxia Gorge from 2002 to 2004.**

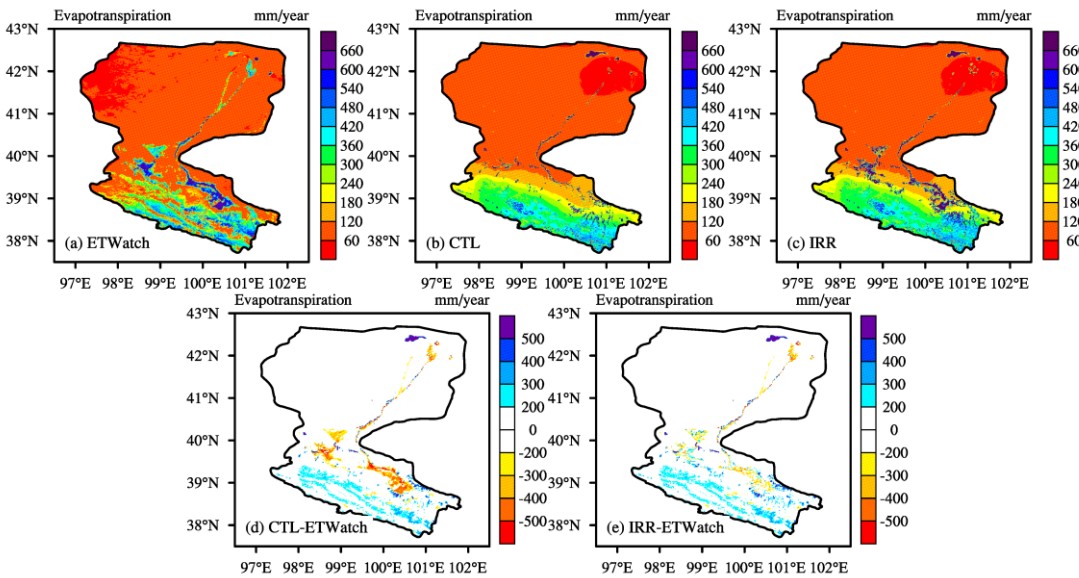

**Figure 6: Climatologic evapotranspiration distribution from 2000 to 2003 over the Heihe River Basin obtained from (a) ETWathch (b) CTL, (c) IRR, (d) CTL-ETWatch, and (e) IRR-ETWatch**

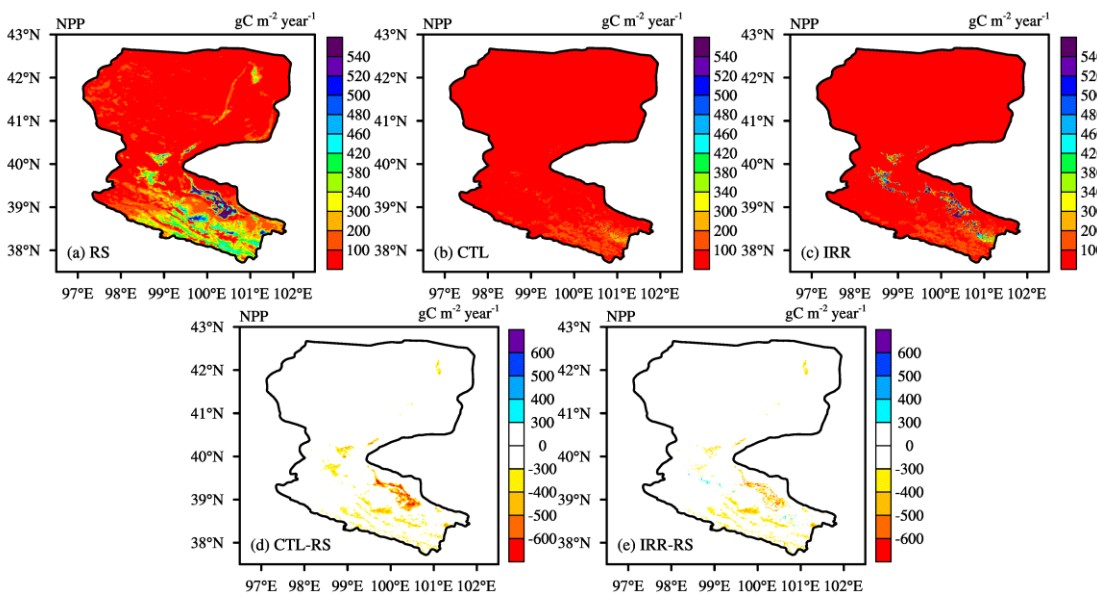

**Figure 7: Climatologic NPP distribution from 1998 to 2002 over Heihe River Basin obtained from (a) remote sensing, (b) CTL, (c) IRR, (d) CTL-remote sensing, and (e) IRR-remote sensing.**

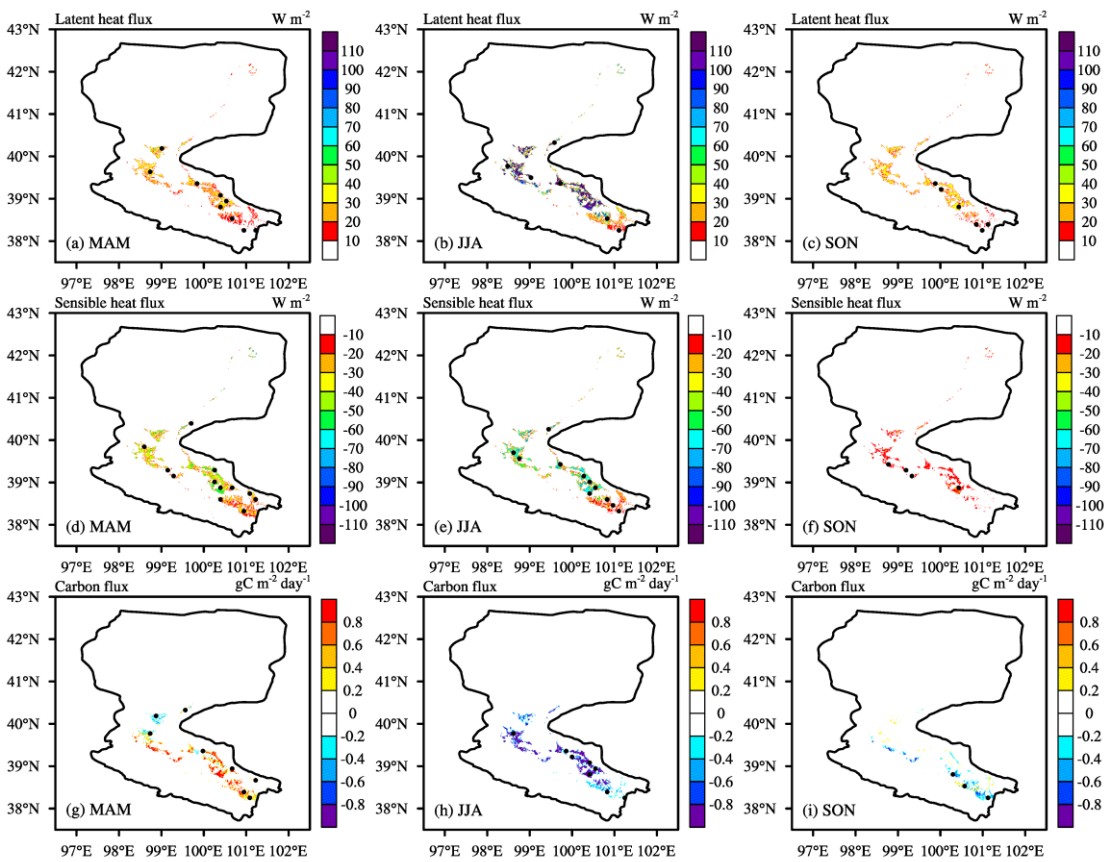

**Figure 8: Climatic spatial pattern of (a-c) latent heat flux, (d-f) sensible heat flux and (g-i) carbon flux differences between IRR and CTL in (a, d, g) spring, (b, e, h) summer and ,(c, f, i) autumn, averaged from 1981 to 2013. The dots represent the area that passed the Student's t-test at a 95% confidence level.**

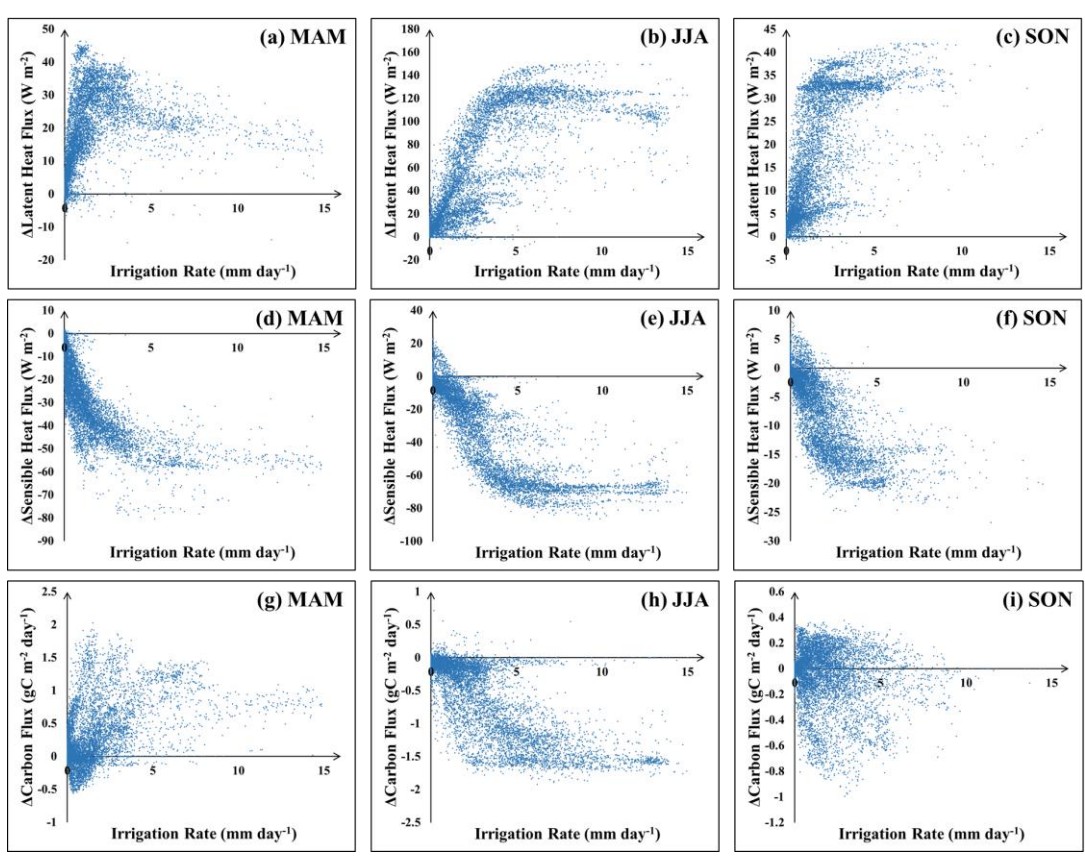

**Figure 9: Differences of (a-c) latent heat flux, (d-f) sensible heat flux and (g-i) carbon flux between IRR and CTL models versus the irrigation rate in (a, d, g) spring, (b, e, h) summer and, (c, f, i) autumn.**

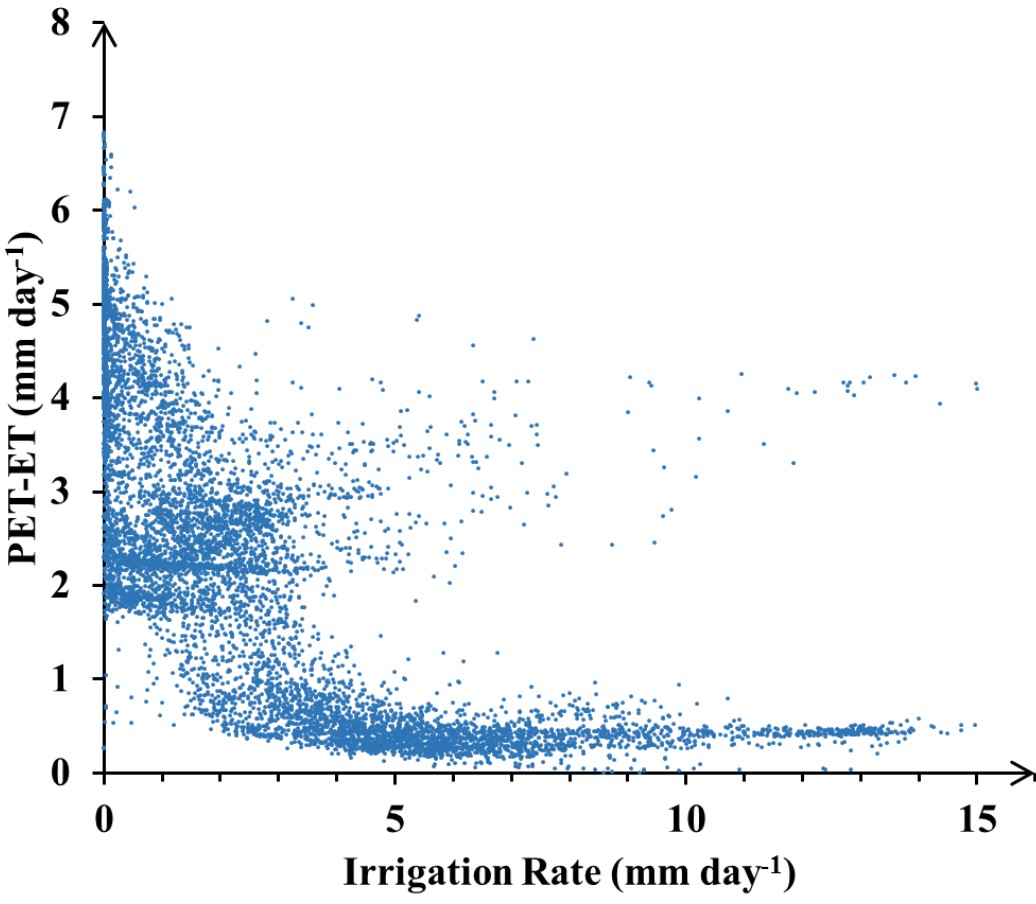

**Figure 10: Differences of potential and actual evapotranspiration versus the irrigation rate in summer.**

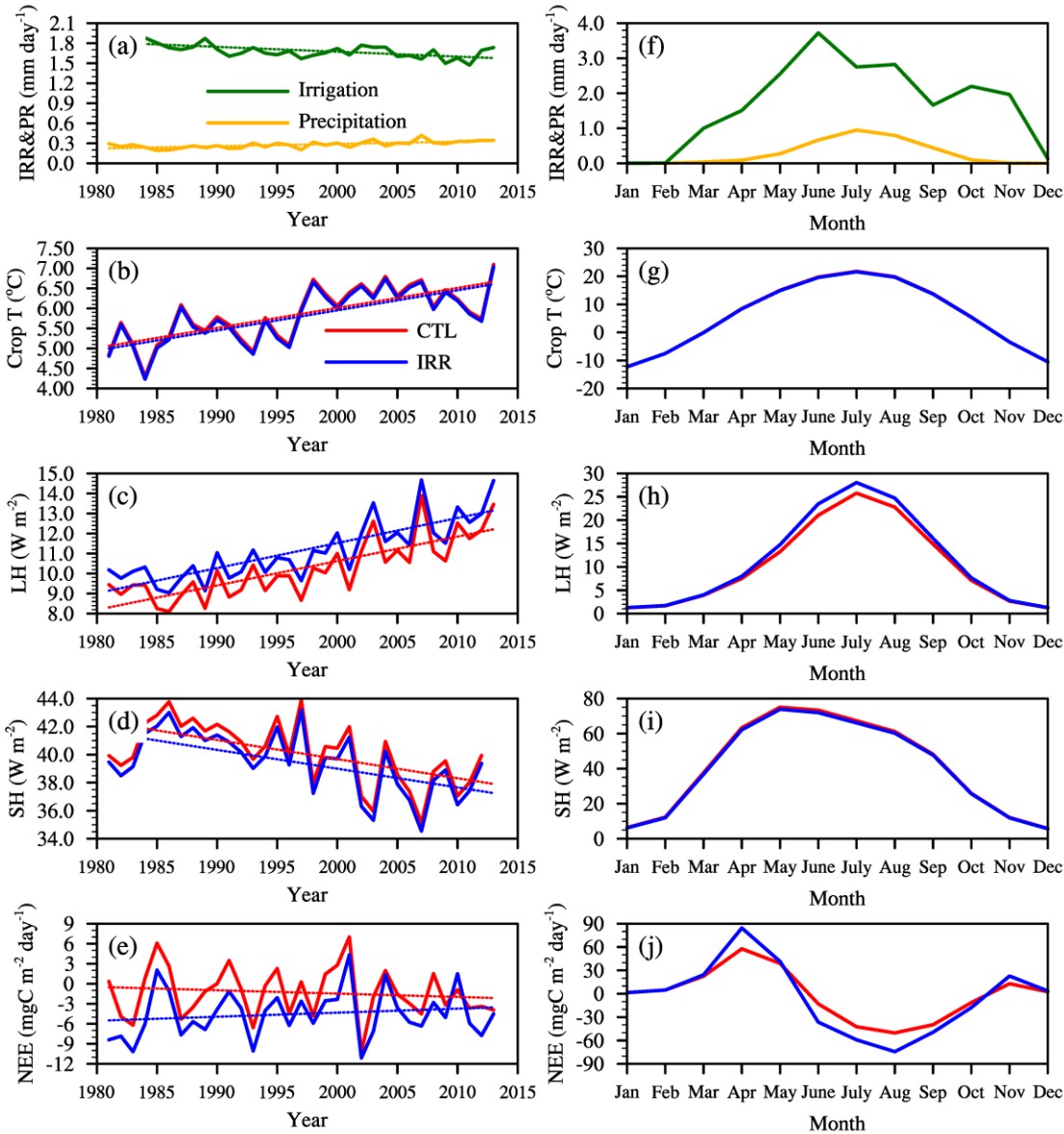

**Figure 11: Time evolution plots of (a-e) annual and (f-j) seasonal (a, f) irrigation and precipitation, (b, g) crop temperature, (c, h) latent heat flux, (d, i) sensible heat flux and, (e, j) carbon heat flux, averaged over all crop grids of the Heihe River Basin.**

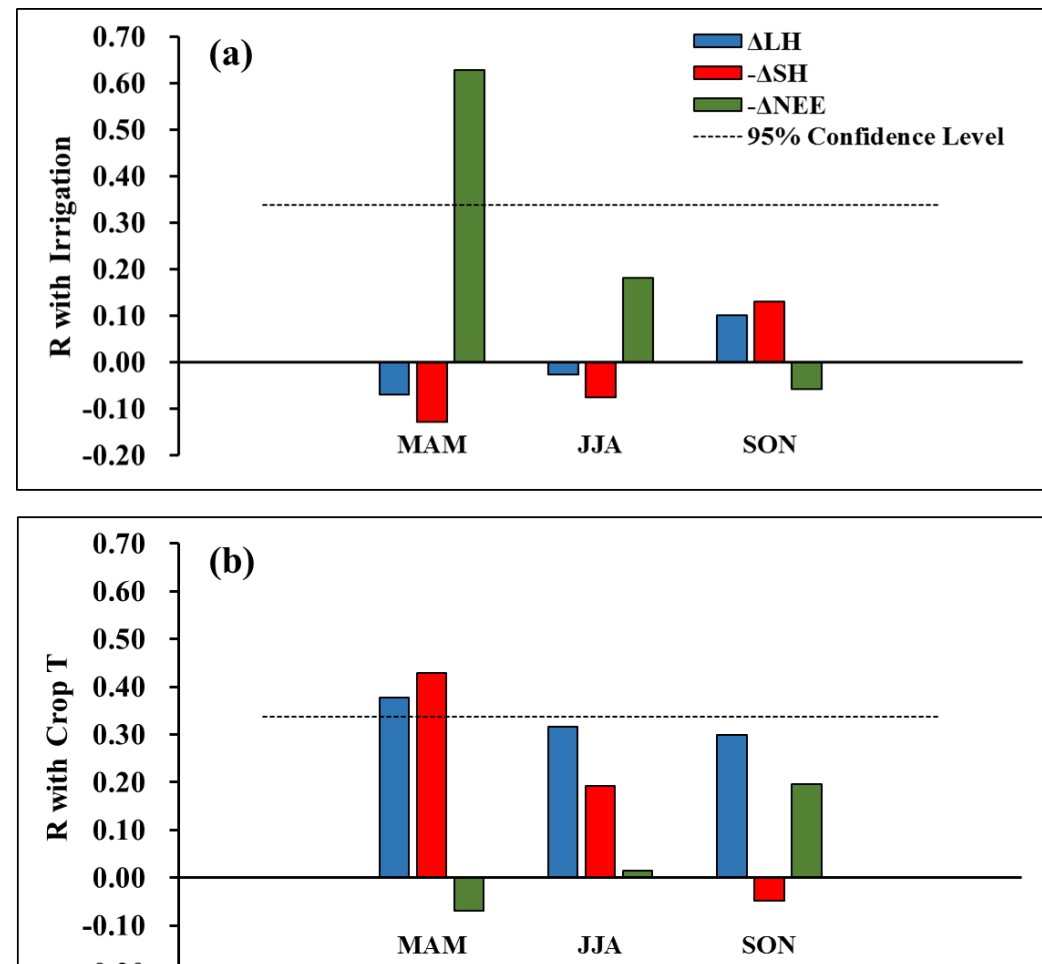

**Figure 12: (a) Temporal correlation coefficients between the irrigation rate and ΔLH, -ΔSH, −ΔNEE over the spring, summer and autumn and, (b) temporal correlation coefficients between the crop temperature and ΔLH, -ΔSH, −ΔNEE over the spring, summer and autumn.**