# Peer review of "Seasonal effects of irrigation on land-atmosphere latent heat, sensible heat and carbon fluxes in semi-arid basin"

_Earth System Dynamics, 2016_

## Referee Comment (RC1) · G. Leng (Referee) · 3 Dec 2016

This study investigate irrigation effects on carbon, water and energy fluxes in an agricultural region in China using CLM4.5. The value of this study lies in applying the high-resolution modeling framework for investigating irrigation effects on carbon flux, in addition to the well documented irrigation effects on water and energy balance components. The paper is well-written and concise. In principal, I am in favor of publication. However, I have several comments that should be addressed before publication. Please find my detailed comments below.

1. Irrigation is prescribed at fixed rates or depend on crop water stress? Please clarify and add discussions on the advantage and disadvantage of the two approach. In addi-

[Figure]

tion, brief introduction on the parameterization of groundwater withdrawals are required in the methodology section, although it was well documented in published papers.

2. Could you please show the spatial pattern of crops considered in this study? Is irrigation treated the same way for the three crops? That is, how irrigation water is determined, abstracted and applied for each of the three crops?

3. Is irrigation efficiency accounted in the experiment? When water is supplied to ground, I would expect substantial losses to runoff and/or groundwater, which is considered in the model. If so, could you elaborate on this and show the range of estimated irrigation efficiency in the model?

4. Authors stated that one advantage of the irrigation scheme is consideration of groundwater withdrawals. In fact, recent works by Leng et al. has done similar studies with CLM for this topic. I suggest to review Leng et al. explicitly in the introduction and method sections. Leng et al. 2014, 2015 are found in the list but not cited in the text.

5. Authors found a threshold of 5mm/day irrigation rate, above which irrigation effects on LH and SH does not change considerably. This is very interesting. Could you please elaborate on this and add discussions on the underlying mechanisms?

6. Figure 5 and 6 shows the ET and NPP from observations and the simulations, respectively. I would suggest adding subplots on the difference between simulations and observations.

Overall, the results are interesting and also informative. I very much enjoyed reading your paper.

Sincerely, Guoyong Leng
* * *

---

## Referee Comment (RC2) · Anonymous Referee #2 · 25 Dec 2016

The manuscript "Seasonal effects of irrigation on land-atmosphere latent heat, sensible heat and carbon fluxes in semi-arid basin" by Zeng et al. investigated the energy fluxes responses to irrigation using a land surface model (CLM 4.5). This study demonstrate a high resolution modeling case using CLM4.5, which combines lateral groundwater transfer and irrigation effects. It is of great interest to the large scale land surface modeling community. I think it's a well-written manuscript and I recommend it for publication after addressing the following comments.

Given that the irrigation is the key part of this manuscript. I think more information about the irrigation scheme should be described here rather than simply refer to another paper. For example, how does the model differentiate the source of the irrigation water

(i.e. groundwater or river water)? Does the model consider any reservoir in the basin for irrigation purpose? When water was taken from groundwater, how the pumping wells be placed across the grid cells since they may create different drawdown behaviors?

The model validation should also include some comparisons between modeled and observed/naturalized stream flows at multiple gauges. I understand the interested variables are energy fluxes in this paper. However, valid stream flow indicates that the model takes right amount of water from the river for irrigations.

The authors found that 5mm/day of irrigation is a threshold beyond which the LH and SH will remain consistent. The authors may want to discuss the mechanism behind this.

Heihe basin is one of the largest inland river basins in China. Most of the irrigation water is from mountainous glacial or groundwater supply. The manuscript shows that the basin was losing about 2mm of water per year constantly during the past 15 years. This trend may change the future groundwater availabilities for irrigations. The author may want to address this point in their discussion.

---

## Author Comment (AC1) · 30 Dec 2016

We thank Dr. Leng for the helpful comments and suggestions. Please also check the attachments for the revised manuscript and our responses to the comments, thanks much for your comments!

1. Irrigation is prescribed at fixed rates or depend on crop water stress? Please clarify and add discussions on the advantage and disadvantage of the two approach. In addition, brief introduction on the parameterization of groundwater withdrawals are required in the methodology section, although it was well documented in published papers.

Response: The irrigation rate in this study was from an external high-quality irrigation

dataset as Section 3.3 described. As suggested, we clarified and added some discussions on the cons and pros of the two approaches (P7, L10-16) and some descriptions for the parameterization of groundwater withdrawal (P5, L33-P6, L19).

2. Could you please show the spatial pattern of crops considered in this study? Is irrigation treated the same way for the three crops? That is, how irrigation water is determined, abstracted and applied for each of the three crops?

Response: As suggested, the spatial pattern of crops was shown in the Figure 1. In the simulations, we did not consider different treat ways for different crop types. It may be taken into consideration in the future (P12, L32-34).

3. Is irrigation efficiency accounted in the experiment? When water is supplied to ground, I would expect substantial losses to runoff and/or groundwater, which is considered in the model. If so, could you elaborate on this and show the range of estimated irrigation efficiency in the model?

Response: Yes, the water losses to runoff/groundwater would be considered by the CLM4.5 in its runoff and infiltration schemes (P6, L17, 18). As suggested, the spatial patterns of the irrigation efficiency were shown in the figure of the supplement. The efficiency was higher in summer and lower in spring and autumn.

4. Authors stated that one advantage of the irrigation scheme is consideration of groundwater withdrawals. In fact, recent works by Leng et al. has done similar studies with CLM for this topic. I suggest to review Leng et al. explicitly in the introduction and method sections. Leng et al. 2014, 2015 are found in the list but not cited in the text.

Response: As suggested, these citations were added in our manuscript (P3, L9-11; P5, L30-32).

5. Authors found a threshold of 5mm/day irrigation rate, above which irrigation effects on LH and SH does not change considerably. This is very interesting. Could you please elaborate on this and add discussions on the underlying mechanisms?

[Figure]

Response: As suggested, the underlying mechanisms was elaborated in the Section 5 (P12, L13-22).

6. Figure 5 and 6 shows the ET and NPP from observations and the simulations, respectively. I would suggest adding subplots on the difference between simulations and observations.

Response: As suggested, the subplots for the difference between simulations and observations were added in Figure 6 and 7.

Please also note the supplement to this comment:
http://www.earth-syst-dynam-discuss.net/esd-2016-45/esd-2016-45-AC1-supplement.zip

---

## Author Comment (AC2) · 30 Dec 2016

We thank the reviewer for the helpful comments and suggestions. Please also check the attachments for the revised manuscript and our responses to the comments, thanks much for your comments!

1. Given that the irrigation is the key part of this manuscript. I think more information about the irrigation scheme should be described here rather than simply refer to another paper. For example, how does the model differentiate the source of the irrigation water (i.e. groundwater or river water)? Does the model consider any reservoir in the basin for irrigation purpose? When water was taken from groundwater, how the pumping wells be placed across the grid cells since they may create different drawdown

behaviors?

Response: As suggested, more information about the irrigation scheme was described (P5, L33-P6, L19). Currently the model does not consider any reservoir in the basin for irrigation purpose, and the pumping wells are placed on the location where the groundwater is consumed. The reservoirs and places of pumping wells may be considered in the future. The discussion above was added in the manuscript (P12, L33-P13, L2).

2. The model validation should also include some comparisons between modeled and observed/naturalized stream flows at multiple gauges. I understand the interested variables are energy fluxes in this paper. However, valid stream flow indicates that the model takes right amount of water from the river for irrigations.

Response: As suggested, a figure on comparison between modeled and observed stream flows was included. Please refer to Figure 5 and P8, L19-28 in the manuscript.

3. The authors found that 5mm/day of irrigation is a threshold beyond which the LH and SH will remain consistent. The authors may want to discuss the mechanism behind this.

Response: As suggested, a discussion for the mechanism was added in the manuscript (P12, L13-22).

4. Heihe basin is one of the largest inland river basins in China. Most of the irrigation water is from mountainous glacial or groundwater supply. The manuscript shows that the basin was losing about 2mm of water per year constantly during the past 15 years. This trend may change the future groundwater availabilities for irrigations. The author may want to address this point in their discussion.

Response: As suggested, the point of water depletion was addressed in the manuscript (P12, L23-29).

Please also note the supplement to this comment:

http://www.earth-syst-dynam-discuss.net/esd-2016-45/esd-2016-45-AC2-supplement.zip

---

## Author Response (AR1)

State Key Laboratory of Numerical Modelling for Atmospheric Sciences
and Geophysical Fluid Dynamics (LASG)
Institute of Atmospheric Physics, Chinese Academy of Sciences

**Dear editor Prof. Roy:**

**We thank you for the helpful comments and suggestions, which are in plain text below. Our response is in bold text.**

The response to comment 5 from Reviewer 1 and comment 3 from Reviewer 2 is speculative. Please add a quantitative analysis and reference to support the claim that at 5mm/day irrigation, "… the water demand of potential ET is fully satisfied and the controlling factor of actual ET converts from water availability to energy availability."

**Response: As suggested, a quantitative analysis and reference to support the claim that at 5mm/day irrigation the water demand of potential ET is fully satisfied was added in the revised manuscript (P11, L9-18, and Figure 10).**

There are many language issues. I have pointed some out below. Please carefully review the entire manuscript for these errors.
Page 1, line 14: Change "… capacity and viability of …" to "… capacity of …"
Page 1, line 14: Change "…results revealed the effects …" to "…results revealed that the effects …"
Page 1, line 16: Change "…irrigation rate below 5 mm …" to "irrigation rate is below 5 mm …"
The last sentence of the abstract should be removed. The study is not about high-resolution crop models. It distracts from the main objective of the manuscript.
Page 2, line 1: Delete "Water". I think this is a typo.
Page 6, line 1: Change "…. aforementioned appoints" to "… aforementioned points".
Page 7 line 10: Change "… it would not be allowed to interact …" to "… it is not allowed to interact …" or better still "… it does not interact …".
Page 8, line 21: Change "… was got from …" to "… was obtained from …"
Page 12, line 24: Change "…reflect the water losing about 2 mm per year." to " … reflect a water loss of about 2 mm per year."

**Response: As suggested, all the sentences above were corrected after we carefully reviewed the entire manuscript.**

[revised manuscript text omitted]